# Relatives from Hereditary Breast and Ovarian Cancer and Lynch Syndrome Families Forgoing Genetic Testing: Findings from the Swiss CASCADE Cohort

**DOI:** 10.3390/jpm12101740

**Published:** 2022-10-19

**Authors:** Mahesh Sarki, Chang Ming, Monica Aceti, Günther Fink, Souria Aissaoui, Nicole Bürki, Rossella Graffeo, Karl Heinimann, Maria Caiata Zufferey, Christian Monnerat, Manuela Rabaglio, Ursina Zürrer-Härdi, Pierre O. Chappuis, Maria C. Katapodi

**Affiliations:** 1Department of Clinical Research, University of Basel, 4055 Basel, Switzerland; 2Swiss Tropical and Public Health Institute, University of Basel, 4123 Allschwil, Switzerland; 3Breast Center, Cantonal Hospital Fribourg, 1752 Fribourg, Switzerland; 4GENESUPPORT, The Breast Centre, Hirslanden Clinique de Grangettes, 1224 Geneva, Switzerland; 5Women’s Clinic, University Hospital Basel, 4031 Basel, Switzerland; 6EOC, Oncology Institute of Southern Switzerland (IOSI), 6500 Bellinzona, Switzerland; 7Institute for Medical Genetics and Pathology, University Hospital Basel, 4031 Basel, Switzerland; 8Research Group Human Genomics, Department of Biomedicine, University of Basel, 4031 Basel, Switzerland; 9Department of Business Economics, Health and Social Care, University of Applied Sciences and Arts of Southern Switzerland, 6928 Manno, Switzerland; 10Department of Medical Oncology, Hospital of Jura, 2800 Delemont, Switzerland; 11Department of Medical Oncology, Inselspital, Bern University Hospital, 3010 Bern, Switzerland; 12Department of Medical Oncology, Cantonal Hospital Winterthur, 8400 Winterthur, Switzerland; 13Unit of Oncogenetics, Division of Oncology, University Hospitals of Geneva, 1205 Geneva, Switzerland; 14Division of Genetic Medicine, University Hospitals of Geneva, 1205 Geneva, Switzerland

**Keywords:** barriers to genetic care, cascade genetic screening, HBOC, healthcare system, relatives

## Abstract

Cascade genetic testing of relatives from families with pathogenic variants associated with hereditary breast and ovarian cancer (HBOC) or Lynch syndrome (LS) has important implications for cancer prevention. We compared the characteristics of relatives from HBOC or LS families who did not have genetic testing (GT (−) group) with those who had genetic testing (GT (+) group), regardless of the outcome. Self-administered surveys collected cross-sectional data between September 2017 and December 2021 from relatives participating in the CASCADE cohort. We used multivariable logistic regression with LASSO variable selection. Among *n* = 115 relatives who completed the baseline survey, 38% (*n* = 44) were in the GT (−) group. Being male (OR: 2.79, 95% CI: 1.10–7.10) and without a previous cancer diagnosis (OR: 4.47, 95% CI: 1.03–19.42) increased the odds of being untested by almost three times. Individuals from families with fewer tested relatives had 29% higher odds of being untested (OR: 0.71, 95% CI: 0.55–0.92). Reasons for forgoing cascade testing were: lack of provider recommendation, lack of time and interest in testing, being afraid of discrimination, and high out-of-pocket costs. Multilevel interventions designed to increase awareness about clinical implications of HBOC and LS in males, referrals from non-specialists, and support for testing multiple family members could improve the uptake of cascade testing.

## 1. Introduction

The US Centers for Disease Control and Prevention classify hereditary breast and ovarian cancer (HBOC) and Lynch syndrome (LS) as Tier 1 genetic conditions. These two syndromes follow an autosomal dominant mode of inheritance, are easily identified through genetic testing, and are actionable [1]. Switzerland has nearly 44,000 new annual cancer cases, with breast, prostate, and colorectal cancers contributing about 33% to all incident cases [2,3]. About 3–15% of incident cases are hereditary in nature [4,5]. Screening unaffected relatives for the familial pathogenic variant is almost 100% accurate and offers the greatest health benefits, particularly to younger individuals, by enabling the timely management of cancer risk [6]. Bilateral prophylactic mastectomy and salpingo-oophorectomy reduce breast and ovarian cancer risk by 90–95% for carriers of HBOC-associated variants, while colonoscopy significantly reduces morbidity and mortality of colorectal cancer associated with LS [7,8]. Despite the potential benefits of genetic testing, rates of cascade testing in relatives from HBOC or LS families have consistently been reported to be less than 50% [9]. The most common predictors of genetic testing are higher personal or familial cancer risk [10,11], followed by provider referrals [11]. Being younger [12,13], having children [14], being female [15], having a higher educational level [13], and being less concerned about the psychological effects of testing [16] are additional predictors of genetic testing. Family support [17,18] and positive experiences with the healthcare system [19] also predict the uptake of genetic testing.

Despite the large number of studies examining the predictors of genetic testing, little is known about relatives forgoing cascade testing even though they are aware of the pathogenic variant running in the family. Studies focusing on untested relatives from HBOC or LS families report that those who opted out of testing appear to be unaware of the benefits of testing [20] and more concerned about negative consequences [21], such as insurance coverage and discrimination [16,20], have higher competing family and social obligations [21], and more difficulty accessing genetic clinics [21]. Several initiatives and trials that involved offering full [22] or partially cost-free testing [23,24], online invitations [23], mailing of saliva kits to relatives following telephone counseling [25], and pre-testing counseling to the entire family [26] did not increase the rates of cascade screening above what has been reported in observational studies. Moreover, most studies report an uptake of genetic testing primarily among first-degree relatives (FDR), who are more likely to be invited and to accept testing [27]. The aforementioned trials also report rates of cascade testing based on responses from index cases, which can be erroneous due to recall and social desirability bias [28,29]. On the other hand, information about cascade testing extracted from medical records cannot exclude the possibility that relatives could have been tested at a different facility [16]. Finally, the above studies have been conducted primarily in the US [23,25,28,29], with fewer conducted in Europe [21,26]. Findings may not apply to other countries with different social and healthcare system structures and cannot provide insights on relatives’ reasons for forgoing cascade testing.

In Switzerland, although the prevalence of HBOC and LS is comparable with other Western countries [30,31], only 11% of breast cancer patients and 25% of cases with a strong family history have genetic testing [32]. Anecdotal data suggest that the uptake of genetic testing is even lower for LS-associated cancer patients. Switzerland has one of the largest healthcare spendings in Europe [33], but barriers to cascade testing among relatives from HBOC and LS families remain poorly understood. This study aims to address this gap. The specific aims were to compare individual, family, and healthcare system characteristics among untested and tested relatives, and to examine reasons for forgoing genetic testing among relatives from known HBOC or LS families.

## 2. Materials and Methods

This study is based on the Swiss CASCADE, a family-based open-ended cohort (NCT03124212) [34]. The protocol has been approved by all appropriate ethics committees (BASEC 2016-02052). CASCADE targets families harboring germline pathogenic variants associated with HBOC or LS and uses surveys to assess cascade testing in relatives and cancer surveillance. Eligible are HBOC and LS index cases and their FDR, second-degree (SDR), and third-degree relatives (TDR) who are ≥18 years old, live in Switzerland, and can complete a survey in either German, French, Italian, or English. The HBOC sample includes families concerned only with *BRCA1* and *BRCA2* pathogenic variants because panel testing was not introduced at the same time at all participating centers. Excluded are individuals diagnosed with variants of unknown significance, non-blood relatives, and critically ill patients unable to provide written consent. Details about the cohort have been previously published [35]. For this analysis, we focused on comparing relatives who did not have genetic testing (GT (−) group) with those who had testing (GT (+) group). Data were collected between September 2017 and December 2021.

HBOC and LS index cases were recruited from eight oncology and genetic testing centers in three linguistic regions of Switzerland. The enrolled index cases received recruitment materials to pass on to relatives whom they were willing to invite to the cohort. The identity of relatives was unknown to the research team; only their gender and biological relationship to the index case was known. Relatives willing to participate returned a signed consent form, revealing their identity and contact information. If the research team did not receive a response from relatives (signed consent or refusal), the index cases were sent one reminder. There were no further attempts to recruit relatives in order to avoid burdening the index case and to avoid potential family conflicts. Relatives agreeing to participate received a baseline survey. Relatives who did not return their survey received one reminder after six weeks, and those who did not respond were considered lost to follow-up. Relatives who submitted baseline data were asked to identify further relatives to invite to the study. Those who tested negative for the familial pathogenic variant, i.e., true negatives, were instructed to not invite their offspring because cascade testing does not apply to them. This additional step (relatives inviting more relatives) applies multiple contact pathways within each family to invite as many at-risk individuals as possible. Respondents received no financial incentive to participate in the cohort. Every January, one respondent was randomly chosen to receive a CHF 300 gift card (approximately USD 310).

At the time of enrollment, relatives were asked if they ever had genetic testing for HBOC or LS and their answer was recorded as “Yes” or “No”. Data on individual characteristics included age, gender, education, personal history of cancer, and perceived cancer risk. Perceived risk was assessed by asking respondents to rate their chance of getting (another) cancer on a 10-point scale, ranging from 1“Definitely not” to 10 “Definitely will”. Family characteristics included number of adults in the family with a cancer diagnosis, number of relatives in the family potentially eligible for cascade testing, total number of tested relatives in the family, and family support in illness. Family support in illness is a 15-item scale [36], where participants indicated their agreement with statements such as “In our family, when I have a health problem, there is no one to turn to”. Items were scored on a 7-point Likert-scale, ranging from 1 “Never true” to 7 “Always”. An average score was calculated, with higher scores indicating greater family support. Cronbach’s alpha was 0.91 in this study.

Healthcare system characteristics evaluated whether respondents had less than or equal to two providers organizing their care and the type of healthcare provider they had seen most often in the past 12 months (specialist vs. general practitioner). Participants were asked whether high out-of-pocket costs were a barrier to accessing healthcare services, by answering whether there has been a time within the past 12 months that they needed to see a doctor or have a medical test but they could not because of cost. Responses were rated on a 7-point Likert-scale, ranging from 1 “Never” to 7 “All the time”. Responses equal or greater than 5 (“Sometimes”, “Often”, “All the time”) indicated cost-related barriers to accessing care. Finally, participants were asked their perceptions of shared medical decision making. The Shared Decision Making and Patient Involvement is a 4-item scale adapted from the Agency of Healthcare Research and Quality index [37]. Items, such as “Your healthcare provider involved you in decisions about your medical care”, were rated on a 7-point Likert-scale, ranging from 1 “Never true” to 7 “Always true”. An average score was calculated, with higher scores indicating a greater engagement of patients and providers in shared decision making. Cronbach’s alpha was 0.81 in the study.

The GT (−) group was asked to respond to a multiple-choice question indicating reasons for not having genetic testing, whereas the GT (+) group indicated their agreement with 11 commonly reported reasons for having genetic testing on a 7-point Likert-scale, ranging from 1 “Strongly disagree” to 7 “Strongly agree”. Finally, we hypothesized that relatives could be tested either at the testing center of the index case or at a testing center close to their place of residence. We calculated relatives’ distance from the testing center of the index case and the nearest testing center based on residence zip codes and zip codes of certified genetic testing centers in Switzerland with the great circle distance formula [38,39,40].

All descriptive and inferential statistical analyses were performed with the statistical software R. Two proportion z-tests and Fischer’s exact tests were used to compare frequencies and proportions. We used the Shapiro–Wilk test to assess whether continuous variables were normally distributed [41] and the non-parametric Wilcoxon rank sum test for non-normally distributed data, when comparing the two groups. Statistical significance was set at *p* < 0.05. There were less than 5% missing data and there was no systematic correlation between missing variables and any covariates. However, due to the small sample size, we applied multiple imputations. Multiple imputations with a “mice” package in R were performed to impute missing values in age, education level, distance, family support, and shared decision making [42]. Multiple imputations created 10 imputed datasets with 60 iterations. We used multivariate logistic regression and LASSO (Least Absolute Shrinkage and Selection Operator) for variable selection, which has the advantage of adjusting for variable multicollinearity [43]. Variable selection was performed for each imputed dataset using a “glmnet” package in R [44]. A three-fold cross validation was performed to find the optimal value of the tuning parameter lambda [44]. The final model was based on the tuning parameter and included predictors with non-zero coefficients in the fitted LASSO model. A sensitivity analysis examined the effect of the family unit on the outcome variable.

## 3. Results

In total, 402 relatives were invited, of whom 115 had completed the baseline survey by December 2021. We excluded (*n* = 20) who were not living in Switzerland and were not eligible for the study. We estimated the response rate (41.4%) conservatively, by excluding relatives still in the recruitment process (*n* = 45) who had not responded to either the invitation or the reminder(s) at the time this analysis was carried out. The response rate was not significantly different between the two syndromes (42.4% for HBOC versus 34.6% for LS, *p* = 0.36). About 11% of the invited relatives (*n* = 45) actively refused to participate. Common reasons reported in the refusal form were “no time, too busy” and “prefer not to answer”.

The index cases had genetic testing a median of 4 years (1–6 years) prior to enrollment, when the pathogenic variant was identified in the family. The index cases were equally likely to invite female and male relatives, and this was not different between the two syndromes. However, respondents were more likely to be female compared with non-respondents (female 68% versus male 47%). The sex of non-responding relatives (no answer to invitation) was provided by the index cases. The sex of relatives who completed the survey was not different from those who consented but did not provide baseline data (female 30.4% versus male 31.0%, *p* = 0.99). We identified 10 relatives potentially eligible for cascade testing per respondent. According to respondents’ data, on average only 2 out of 10 relatives had cascade screening (Figure 1).

The 115 relatives came from 70 different families. Most were female (68%), from families with HBOC-associated variants (88%) and who had no personal history of cancer (20%) (Table 1). Nearly 62% (*n* = 71) had genetic testing (GT (+) group), while 38% (*n* = 44) did not have genetic testing (GT (−) group). The GT (−) group included a significantly higher proportion of males and a lower proportion of cancer patients compared with the GT (+) group. The median number of tested individuals was lower in families of the GT (−) group compared with the GT (+) group (2 versus 3, *p* < 0.01). Finally, relatives in the GT (−) group were more likely to report that during the past 12 months they had seen most often a non-specialist (family doctor or general practitioner) and/or that more than two healthcare providers had coordinated their care, compared with the GT (+) group.

The proportion of tested relatives was not different by syndrome (60% HBOC vs. 71% LS, *p* = 0.62). The majority (76%) of the GT (+) group were identified as carrying the familial pathogenic variant, with the rest being true negatives. Approximately half (49%) had genetic testing during the past 5 years. The total number of cancers and the number of adults in the family potentially eligible for cascade testing were not statistically different between the GT (−) and the GT (+) groups. The proportions of FDR, SDR, and TDR were not significantly different between the GT (−) and the GT (+) groups and by syndrome.

Logistic regression showed that male relatives had nearly three times the odds (OR: 2.79, 95% CI: 1.10–7.10) of being untested compared with females (Table 2). Individuals without a cancer diagnosis had almost four times the odds of being untested (OR: 4.47, 95% CI: 1.03–19.42). Individuals from families with fewer tested relatives had 29% higher odds of being untested (OR: 0.71, 95%CI: 0.55–0.92). None of the healthcare system characteristics were selected by LASSO as a significant predictor of cascade testing. However, having less than or equal to two healthcare providers coordinating care was retained in the model as a borderline significant predictor of being in the GT (−) group.

Among relatives in the GT (−) group, the most commonly reported reason for not having genetic testing was “No one ever suggested it” both for female and male relatives (Figure 2). Other reasons mentioned by both females and males in the GT (−) group were “I don’t want to/I am not interested in/I have the right not to know”, “I can’t afford out of pocket expenses”, “I am too busy”, “Other life issues that come up are more important”, and “I am afraid the results could be used against me”. However, “I can’t get time off work” and “I don’t know where to go for these services” were mentioned only by females, whereas “My doctor said I don’t need it” and “A family member tested negative” were mentioned only by males as reasons for not having genetic testing.

Most relatives in the GT (+) group reported that a specialist (i.e., oncologist, gynecologist, gastroenterologist, geneticist or genetic counselor, or surgeon) (40%) or a relative (34%) recommended genetic testing (Figure 3). Significantly fewer were self-referrals (10%) or received a recommendation from a family doctor (5%). Most female and male relatives in the GT (+) group mentioned that they had genetic testing because they wanted to know “…if (they) have a pathogenic variant connected to cancer”, “… more about (their) future cancer risk”, and the “… risk for their children”. However, only females agreed with the statement that “results may change (their) family planning”.

## 4. Discussion

The study examined self-reported rates of cascade testing among relatives from HBOC and LS families. The GT (−) group included a higher proportion of individuals without cancer, while having a cancer diagnosis increased the odds of cascade testing, which is consistent with other studies [11,16,22]. A personal history of cancer brings the individual in contact with the healthcare system, where risk factors are likely to be identified, and prompts referrals for genetic evaluation [45]. Consistent with the existing literature [9], the most common reason for not having genetic testing reported by the GT (−) group was the lack of a recommendation from a healthcare provider. The GT (−) group was also more likely to report that two or more healthcare providers coordinated their care. Although follow-up medical visits can facilitate cascade testing [46], it is also possible that when the individual receives care from multiple healthcare providers, none assumes responsibility specifically for cascade testing.

The GT (−) group was also more likely to report receiving care from non-specialists. Specialists are more likely to identify individuals who need genetic evaluation compared with general practitioners [47]. Non-specialists often lack genetic knowledge, do not perform genetic risk assessments, and are less likely to order genetic tests [48]. Our findings suggest that educational programs targeting non-specialists in using family history as a risk assessment tool may significantly increase referrals for genetic evaluation [49], bridge the gap between primary and specialized care, and eventually promote cascade testing [50]. Others also highlight the need for engaging providers from multiple specialties in promoting cascade testing. For example, the Dutch have successfully achieved nationwide screening for familial hypercholesterolemia (FH), another Tier 1 genetic condition, with nearly 90% of FH cases having most of their FDR successfully screened through a provider-mediated invitation [51]. The Dutch model provides useful information for the implementation of cascade screening programs in high healthcare-spending countries, including Switzerland, and suggests provider-mediated invitations to HBOC- and LS-cascade testing programs, following mainstream cancer-related genetic testing [52,53].

Although index cases were equally likely to invite male and female relatives to the study, males were significantly less likely to accept this invitation and had higher odds of being untested than females. This finding likely reflects the lack of awareness of the clinical implications of hereditary cancer for males and, specifically, the 40% risk for early onset prostate cancer among male carriers of HBOC-associated variants [28,29]. Interventions targeting males are needed to improve the gender gap for HBOC cascade testing [22,23]. Different modes of genetic counseling and decision aids that have benefited females from HBOC- and LS-harboring families could also be explored for males [54].

The odds of cascade testing increased with the number of tested individuals within each family, suggesting that family members influence and motivate each other for testing [55]. Parity is also consistently reported as a reason for cascade testing [14]. Findings highlight the need for family-based interventions that leverage existing networks for disseminating information and increasing awareness about hereditary cancer. Finally, a few untested relatives mentioned the lack of insurance coverage as a reason for forgoing cascade testing. Basic health insurance in Switzerland does not cover the cost of cascade testing for SDR and TDR [56]. Given that approximately half of the non-invited relatives in the CASCADE cohort are SDR and TDR [35], out-of-pocket costs (about CHF 400 or USD 410) could be a barrier for the approximately 27% of untested SDR and TDR in the study. Additional reasons for foregoing testing, such as not interested, busy, and valuing other life events more than genetic testing, have been reported also by others [16,21,27,29].

The strength of our study is that the uptake of genetic testing is reported directly by relatives, as opposed to obtaining this information from index cases or from medical files, which may be biased or inaccurate [28,29]. It is also possible that the index cases invited relatives with whom they have a better relationship and more open communication, which may also mean more invitations for those who followed through with cascade testing rather than untested relatives. Thus, it is also possible that the rates of cascade testing are significantly lower among non-invited and non-participating relatives. Another limitation of the study is the higher representation of HBOC relatives compared with LS. The HBOC sample includes families concerned only with *BRCA1* and *BRCA2* pathogenic variants, as panel testing was not introduced simultaneously in all participating clinical sites. To avoid subject burden, only one reminder was sent before a relative was considered lost to follow-up, which could have influenced the participation of relatives in the study. The cross-sectional data presented here cannot capture dynamic characteristics that may influence cascade testing. The small sample size of the non-tested relatives affects the statistical power and generalizability of the findings. A considerable number of invited relatives, especially males, did not participate in the study. Non-respondents may be different regarding additional demographic and clinical characteristics and health-seeking behaviors beyond what we assessed. Information for the cancer and genetic testing status of non-responders was reported by the index cases and may be inaccurate. Some of these limitations can be overcome with time, as CASCADE is an open-ended cohort.

## 5. Conclusions

Multiple factors influence decisions to forgo cascade testing, therefore multifactorial interventions are needed. Interventions should focus on increasing the engagement of non-specialists in genetic evaluation by assessing family history, facilitating referrals, increasing coordination of healthcare services with emphasis on creating stronger ties between primary care and genetic services, and implementing family-based interventions that emphasize the clinical implications of hereditary cancer syndromes for males. Evidence is needed to support the acceptability of these approaches among patients and clinicians and the feasibility of implementing necessary changes in existing infrastructures.

## Figures and Tables

**Figure 1 jpm-12-01740-f001:**
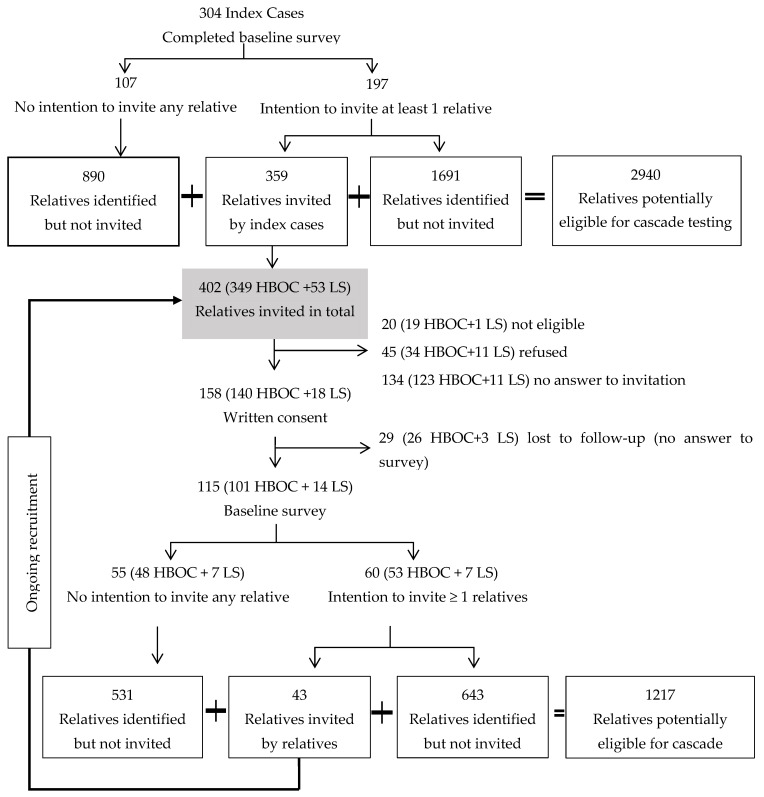
Recruitment of relatives from hereditary breast and ovarian cancer (HBOC) and Lynch syndrome (LS) families.

**Figure 2 jpm-12-01740-f002:**
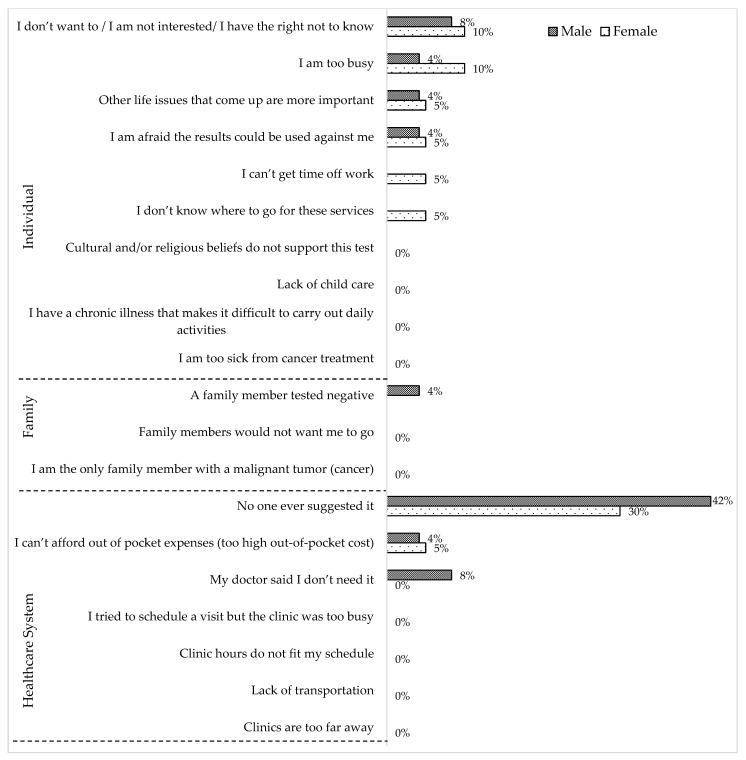
Reasons for not having genetic testing reported by the GT (−) group (*n* = 44) according to sex (female = 20 and male = 24).

**Figure 3 jpm-12-01740-f003:**
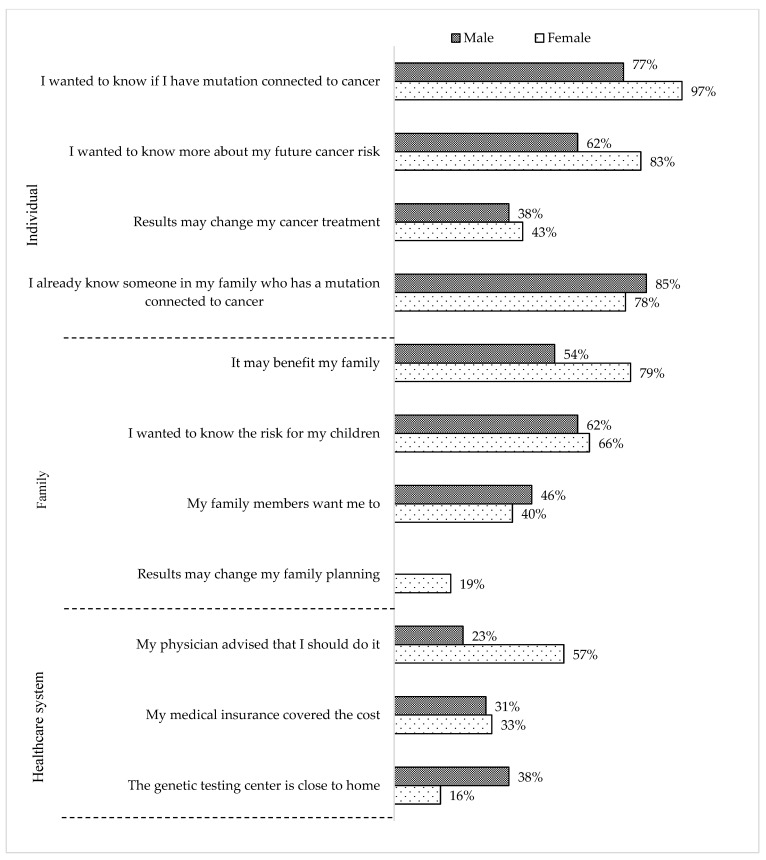
Reasons for having genetic testing reported from the GT (+) group (*n* = 71) according to sex (female = 58 and male = 13).

**Table 1 jpm-12-01740-t001:** Comparison of individual, family, and healthcare system characteristics between the GT (−) and the GT (+) groups.

		GT (−) *n* = 44	GT (+) *n* = 71	*p* Value
**Individual Characteristics**				
Demographics	Age, years	49 (33.8–67.3)	47 (35.3–63.8)	0.26
	Male, *n* (%)	24 (54.5)	13 (18.3)	**<0.01 ^a^**
	Female, *n* (%)	20 (45.5)	58 (81.7)	**<0.01**
	Index case being male, *n* (%)	12 (27.3)	8 (11.3)	**0.05**
	HBOC, *n* (%)	40 (90.9)	61(85.9)	0.62
	LS, *n* (%)	4 (9.1)	10 (14.1)	0.57 ^b^
	More than 12 years of education, *n* (%)	25 (56.8)	45 (63.4)	0.61
	Employed, *n* (%)	27 (61.4)	43 (60.6)	0.99
	Married or living with a partner, *n* (%)	33 (75.0)	51 (71.8)	0.88
Clinical	Personal history of cancer, *n* (%)	3 (6.8)	20 (28.2)	**0.03 ^b^**
Costs	Out-of-pocket costs as barrier to medical care, *n* (%)	1 (2.3)	1 (1.4)	0.99 ^b^
Psychological	Perceived risk to get (another) cancer, median (IQR)	6 (4.5–7.0)	7 (5.0–8.0)	0.14 ^c^
Relatives’ degree of relationship with the index case ^d^	FDR, *n* (%)	30 (68.2)	48 (67.6)	0.99
	SDR, *n* (%)	7 (15.9)	6 (8.5)	0.36
	TDR, *n* (%)	5 (11.4)	8 (11.3)	0.99
**Family characteristics**				
Adults in the family with cancer	Number of adults in the family with cancer, median (IQR)	1 (1–1.3)	1 (1–2)	0.09 ^a^
Adult relatives in the family potentially eligible for cascade testing	Number of relatives in the family potentially eligible for cascade testing, median (IQR)	9.5 (5.8–14.0)	10 (6–17)	0.29 ^a^
Relatives willing to invite more relatives to the cohort	Number of relatives willing to invite more relatives to the cohort, median (IQR)	0 (0–0)	1 (0–4)	**<0.01 ^a^**
Adult relatives tested in the family	Number of relatives tested in the family, median (IQR)	2 (1–3)	3 (2–6)	**<0.01 ^a^**
Family support	Family support in illness, median (IQR)	6.3 (5.8–6.6)	6.1 (5.5–6.6)	0.61 ^b^
**Healthcare system characteristics**				
	Coordination of care by specialist, *n* (%)	12 (27.3)	36 (50.7)	**0.02**
	≤2 healthcare providers organize care, *n* (%)	36 (81.8)	68 (95.8)	**0.03**
	Shared decision making and patient involvement, median (IQR)	5.5 (3.5–6.9)	6.0 (4.0–7.0)	0.38 ^c^
	Distance to the nearest genetic testing center, km, median (IQR)	10.4 (5.6–15.6.)	8.9 (3.2–21.8)	0.85 ^c^
	Distance to the index’s case testing center, km, median (IQR)	27.5 (14.9–69.1)	21.9 (3.5–63.2)	0.32 ^c^

Bold: statistically significant (<0.05); ^a^ two proportion z-test, ^b^ Wilcoxon rank sum test, ^c^ Fisher’s exact test; ^d^ for 9 relatives in the GT (+) group and 2 relatives in the GT (−) group, information about their degree of relationship to the index case was missing; IQR—inter-quartile range; FDR—first-degree relatives; SDR—second-degree relatives; TDR—third-degree relatives; km—kilometers.

**Table 2 jpm-12-01740-t002:** Odds of being in the GT (−) group versus the GT (+) group.

		OR ^a^	95% CI	Std. Error	Statistic	*p*
**Individual Characteristics**						
Demographic	Male (ref: female)	**2.79**	**1.10–7.10**	0.48	2.15	**0.034**
Clinical	No cancer diagnosis (ref: having cancer diagnosis)	**4.47**	**1.03–19.42**	0.75	2.00	**0.048**
**Family characteristics**						
Total number of tested relatives in the family		**0.71**	**0.55–0.92**	0.13	−2.63	**<0.01**
**Healthcare system characteristics**						
≤2 healthcare providers organize care (ref: >2 healthcare providers)		0.23	0.04–1.14	0.82	−1.80	0.074

Bold: statistically significant; ^a^ odds ratio based on logistic regression fitted model with LASSO variable selection; CI—confidence interval; Std. error—standard error.

## Data Availability

The CASCADE Consortium is open to collaboration with national and international researchers. Interested parties can contact the PI to discuss project ideas and access to data. Decisions are made in collaboration with the Scientific Board. Templates for data requests are available at https://swisscascade.ch/en/research-project-data-request/ (accessed on 15 June 2022).

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
