# Peer review of "Relatives from Hereditary Breast and Ovarian Cancer and Lynch Syndrome Families Forgoing Genetic Testing: Findings from the Swiss CASCADE Cohort"

_jpm, 2022, doi:10.3390/jpm12101740_

Round 1

Reviewer 1 Report

This manuscript entitled Relatives from hereditary breast and ovarian cancer and Lynch syndrome families forgoing genetic testing: Findings from the Swiss CASCADE cohort aims to compare characteristics between relatives affected by breast and ovarian cancer (HBOC) and Lynch syndrome (LS) untested. The authors compared relatives (GT(-) group) and relatives who had genetic testing regardless of the testing outcome (GT(+) group). This paper presents important information for implementing cascade screening programs for hereditary breast and ovarian cancer and Lynch syndrome-affected families and finds reasons for waiving genetic testing among relatives from known HBOC or LS families.

The manuscript is well written. In my opinion, some points of the paper need to be revised.

The authors wrote in the Material and Methods section: “Non-parametric Wilcoxon rank sum test was applied non-normally distributed data for comparing the two groups”; p. 4 lines 121-122. Considering the results and Table 1 footnote, they applied the Mann-Whitney U test; please clarify. Additionally, to this point, did the authors test the distribution for the Likert scale or take the variables as ordinal? If it is the first case, they should indicate the test for distribution assessment; if the second, the sentence should be rewritten to point to the proper test for variables in the ordinal scale.

Firstly, I am not sure if Figure 1 is necessary for the paper; secondly, the paragraph below Figure 1, from lines 182 and 189 (The GT(+) and GT(-) authors in a highly complex way describing the subgrouping. The authors described, e.g., ‘Among the 115 relatives, 62% were in the GT(+) group (n=71) and 38% were in the GT(-) group (n=41) (Table 1)’. I am a little confused: 71+41=112, not 115. Additionally, in Table 1, we see n=71 and n=44, then the number summarizes to 115.

Table 2, because of typography, is hard to read. Additionally, regarding the regression analysis, the odds ratio is more of a positive meaning, then in lines 203 and 204 “odds ratios of being untested” sounds a little in surprising way. I know that you counted the odds ratio, and the relationship between odd and risk ratio is complicated, but could you also show the risk ratio? Additionally, we should not consider OR if both the OR or/and CI are below 1 (line 205). It is just a question for me; are the ORs logistic regression estimates? There was some designation missing for the test used (e.g., b), which refers to missing designation in the table.

The conclusions could be modified to support the results presented in the paper in a closer way.

The authors should take care of the proper spelling of the words. I am not an English native, but the authors mix British and American English. Thus it is hard to determine if the paper is either in one or second version, e.g., on pages 2 and 3, line 59 counseling (American) and line 60 counselling (non-American); p. 10, l. 227 gynecologists (American), l. 228 counselling (non-American). There are minor flaws in the whole manuscript (singular/plural, redundant or missing articles, hyphenation, etc.).

Overall the conclusions are interesting and, in my opinion, important in the aspect of a population-based study. Additionally, findings have significant implications for cascade screening programs.

Author Response

Dear Reviewer 1,

Please see the attachment with a point by point response to your comments.

Reviewer 2 Report

Overall, I think this is a very meaningful topic. This study focused on the reasons why relatives of HBOC and LS patients accepted or refused genetic testing. However, there are still some improvements to be made in the manuscript, as follows:

Abstract:

GT should have a full name when it first appeared.

Keywords:

In my opinion, the author should delete “family characteristics”, and add “breast and cancer”, “ovarian cancer’’ and “Lynch syndrome”. Meanwhile, “untested relatives” should be changed into “relatives”.

Introduction:

1.The significance of genetic testing: Prophylactic mastectomy can be considered for breast cancer, but what about cervical and colorectal cancer? For some relatives, is this just adding to needless fear? It is suggested to add the explanation of the significance of genetic testing.

2.In the first paragraph, it is suggested to supplement the incidence of HBOC and LS.

3.In the first paragraph, the sentence “Prevention and risk management strategies reduce the risk of HBOC-associated breast cancer by 90-95% [2]. Similarly, colonoscopy significantly reduces the burden of LS-associated colorectal cancer [3].” First, what is the difference between prevention strategies and risk management strategies? Secondly, how do “the prevention and risk management strategies of HBOC and colonoscopy of LS” relate to the topic of this study (GT)?

4.In the first paragraph, the sentence “The most common predictors of genetic testing uptake are higher personal or familial cancer risk [6, 7], followed by provider referrals [7]. Being younger [8, 9], having children [10], being female [11], having higher educational level [9], and being less concerned about psychological effects of testing [12] are additional predictors of cascade screening. Finally, family support [13, 14] and positive experiences with the healthcare system [15] facilitate uptake of genetic testing.” The last one is said to be a facilitator, but the first few are only said to be a predictor, not a facilitator or a hindrance.

5.In the second paragraph, “Moreover, the observed effect in uptake of genetic testing involved primarily first-degree relatives (FDR), who are more likely to be invited and to accept testing [23].” What do you mean by this sentence?

6.In the second paragraph, the sentence “The aforementioned trials also report cascade testing rates either from index cases and are subject to recall and social desirability bias [24, 25], or from medical records and cannot exclude the possibility that relatives were tested at a different facility [12]” need to be rewritten.

7.In the third paragraph, according to the following passage structure, do you consider putting “to compare individual,family, and healthcare system characteristics among untested and tested relatives” in front of “to find out reasons for forgoing genetic testing among relatives from known HBOC or LS families”?

Materials and Methods

1.In the first paragraph, “Eligible are HBOC and LS index cases and their FDR, second (SDR)-, third degree relatives (TDR)”. However, the first sentence of the "Discussion" section said that the data of this study were directly reported by relatives, not index cases. So did the study include patients with HBOC and LS?

2.It is suggested to supplement the connotation of FDR, SDR and TDR.

3.In the second paragraph, “Relatives who submitted baseline data were asked to identify further relatives to invite to the study.” Should it be changed to “Relatives who submit baseline data and test positive for familial pathogenic variants were asked to identify further relatives to invite to the study”?

4.Is each subject tested for familial pathogenic variants? At the time of enrollment, some have undergone genetic testing and some have not. For those who have not, where should they be tested? Who will bear the cost?

5.In the third paragraph, “Family characteristics included number of adults with a cancer diagnosis in the family”. Why limit "adults" and why not "childs"?

6.In the forth paragraph, “≤2 providers organize their care”, followed by “having more than two healthcare providers coordinating care”, which is inconsistent. And why is it bounded by two?

7.In the fifth paragraph, “Finally, the GT(-) group was asked to respond to a multiple-choice question indicating reasons for not having genetic testing, whereas the GT(+) group indicated their agreement with 11 commonly reported reasons for having genetic testing on a 7-point Likert-scale.” How were these reasons determined? Is there any evidence? The accuracy and comprehensiveness of these reasons are alarming.

Results:

Table 1:

1.“Clinical Personal history of cancer”. Are they common cancer or hereditary cancer?

2.“Costs Out of pocket cost as barrier to medical care”. How to judge it?

Figure 2:

3.“Clinic hours do not fit my schedule” appears twice.

Discussion:

1.In the first paragraph, “However, it is also possible that index cases invited relatives with whom they have a better relationship and more open communication, which may also mean more invitations for relatives who followed through with cascade testing rather than untested relatives. Thus, it is also possible that rates of cascade testing are significantly lower among non-invited and non-participating relatives.” Should this sentence be placed in the limitation section?

2.In the six paragraph, does “Information for non-respondents’ cancer and genetic testing status was reported by the index cases and may be prone to bias” contradict the first sentence “The strength of our study is that uptake of testing is reported directly by relatives, as opposed to obtaining this information from index cases or from medical files, which may be biased or inaccurate” in the Discussion section? If some general information was provided by index cases, what was the reason for the refusal of these people to answer? It is recommended to indicate the sources of the different data types in the "Methods" section.

Author Response

Dear Reviewer 2,

Please see the attachment for a point by point response to your comments.

Round 2

Reviewer 2 Report

I think this version is muh better than the original one.